# Comparative Analysis of Swine Antibody Responses following Vaccination with Live-Attenuated and Killed African Swine Fever Virus Vaccines

**DOI:** 10.3390/vaccines11111687

**Published:** 2023-11-03

**Authors:** Hung Q. Luong, Huong T. L. Lai, Lam Q. Truong, The N. Nguyen, Hanh D. Vu, Hoa T. Nguyen, Lan T. Nguyen, Trang H. Pham, D. Scott McVey, Hiep L. X. Vu

**Affiliations:** 1Nebraska Center for Virology, University of Nebraska-Lincoln, Lincoln, NE 68583, USA; qluong4@unl.edu (H.Q.L.); tnguyen175@huskers.unl.edu (T.N.N.); 2School of Veterinary Medicine and Biomedical Sciences, University of Nebraska-Lincoln, Lincoln, NE 68583, USA; dmcvey2@unl.edu; 3Faculty of Veterinary Medicine, Vietnam National University of Agriculture, Hanoi 100000, Vietnam; ltlhuong@vnua.edu.vn (H.T.L.L.); tqlam@vnua.edu.vn (L.Q.T.); vdhanh@vnua.edu.vn (H.D.V.); hoanguyenty@vnua.edu.vn (H.T.N.); nguyenlan@vnua.edu.vn (L.T.N.); htrang2910@gmail.com (T.H.P.); 4Department of Animal Science, University of Nebraska-Lincoln, Lincoln, NE 68583, USA

**Keywords:** African swine fever virus, live attenuated vaccine, inactivated vaccine, antibody profile, humoral immunity, luciferase-based immunoprecipitation system

## Abstract

African swine fever virus (ASFV) is circulating in many swine-producing countries, causing significant economic losses. It is observed that pigs experimentally vaccinated with a live-attenuated virus (LAV) but not a killed virus (KV) vaccine develop solid homologous protective immunity. The objective of this study was to comparatively analyze antibody profiles between pigs vaccinated with an LAV vaccine and those vaccinated with a KV vaccine to identify potential markers of vaccine-induced protection. Thirty ASFV seronegative pigs were divided into three groups: Group 1 received a single dose of an experimental LAV, Group 2 received two doses of an experimental KV vaccine, and Group 3 was kept as a non-vaccinated (NV) control. At 42 days post-vaccination, all pigs were challenged with the parental virulent ASFV strain and monitored for 21 days. All pigs vaccinated with the LAV vaccine survived the challenge. In contrast, eight pigs from the KV group and seven pigs from the NV group died within 14 days post-challenge. Serum samples collected on 41 days post-vaccination were analyzed for their reactivity against a panel of 29 viral structural proteins. The sera of pigs from the LAV group exhibited a strong antibody reactivity against various viral structural proteins, while the sera of pigs in the KV group only displayed weak antibody reactivity against the inner envelope (p32, p54, p12). There was a negative correlation between the intensity of antibody reactivity against five ASFV antigens, namely p12, p14, p15, p32, and pD205R, and the viral DNA titers in the blood of animals after the challenge infection. Thus, antibody reactivities against these five antigens warrant further evaluation as potential indicators of vaccine-induced protection.

## 1. Introduction

African swine fever virus (ASFV) is the causative agent of a hemorrhagic disease in pigs, with the mortality rate often approaching 100%. Over the past few years, the virus has spread to many countries in Asia, Europe, and Central America, causing substantial economic losses [1]. ASFV is a large double-stranded DNA virus belonging to the genus *Asfirvirus,* in the family *Asfarviridae*. The viral genome varies between 170 and 193 kbp and contains between 150 and 170 open reading frames (ORFs) [2]. Over 100 viral proteins have been identified in ASFV-infected cells [3]. The ASFV virion has a complex structure, consisting of a nucleoid, a core shell, and a capsid surrounded by inner and outer membranes, respectively [4]. At least 68 virus-encoded proteins have been identified in the virus particles, half of which have no known functions [5].

Under experimental conditions, pigs infected with a highly virulent ASFV strain often succumb to the infection before they can mount a detectable immune response to the virus [6,7]. Conversely, pigs infected with naturally low or moderately virulent ASFV strains typically survive and become resistant to reinfection by closely related virulent strains [8,9]. In pigs infected with low or moderately virulent ASFV strains, virus-specific antibodies can be detected between 7 and 14 days post-infection (dpi) [10,11]. The passive transfer of hyperimmune antibodies to naïve pigs resulted in protection against lethal challenge infection with a virulent ASFV strain [12]. Furthermore, there is a clear positive correlation between antibody titers and protection outcomes in pigs vaccinated with an ASF live attenuated vaccine (LAV) candidate and subsequently challenged with a virulent ASFV strain [13]. Thus, antibodies are an essential component of protective immunity against ASFV. 

While the viral genome encodes for up to 170 proteins, only a small number of these proteins have been characterized for their ability to induce antibody responses in infected pigs. Through screening the viral cDNA library using a polyclonal antiserum obtained from one pig surviving an infection with a virulent ASFV strain, 14 immunogenic proteins were identified [14]. The serological responses to these proteins can be divided into three groups: strong response (E183L/p54, K205R/‘unassigned’, A104R/histone-like, and B602L/‘unassigned’), intermediate response (B646L/p72, CP204L/p32, CP312R/‘unassigned’, NP419L/DNA ligase, and F334L/ribonucleotide reductase), and low antibody responses (K169R/thymidine kinase, K78R/p10, and C44L/‘unassigned’) [10]. The identification and detailed characterization of swine antibody responses to these 14 immunogenic proteins have provided insights for the development of serological diagnostic tests [15,16,17]. 

Extensive efforts have been dedicated to the development of an effective vaccine against ASFV (reviewed in [18]). Killed virus (KV) vaccines have proven ineffective in providing protection, even when tested against homologous virus strains [19,20,21]. The protective efficacy of ASF subunit vaccines varies considerably. In general, the subunit vaccines can elicit an adaptive immune response in vaccinated pigs, but they often fail to confer protection against lethal challenge with a homologous virulent ASFV strain [22]. The main obstacle to the successful development of an effective subunit vaccine is our limited knowledge of viral antigens capable of inducing protective immunity. Multiple live-attenuated virus (LAV) vaccines have been developed by the targeted removal of virulent genes from the viral genome (reviewed in [23]). Under experimental conditions, these LAV vaccine candidates confer solid protection against lethal challenge with the parental virulent ASFV strains by significantly reducing the viral load in blood, reducing or absence of clinical signs, and 100% survival. In some instances, the LAV candidates have been reported to confer sterilizing immunity [24]. 

Our group has recently developed an LAV vaccine candidate by successively passaging a virulent ASFV strain on primary and immortalized porcine alveolar macrophages (PAMs) for a total of 120 passages [25]. The resulting virus, designated VNUA-ASFV-LAVL2, replicates efficiently in both primary and immortalized PAMs. Under experimental conditions, the VNUA-ASFV-LAVL2 is fully attenuated and confers complete protection against the parental virulent ASFV strain, even at the low immunization dose containing 10^2^ HAD50.

The primary objective of this study was to compare the antibody profiles of pigs vaccinated with experimental LAV and KV vaccines to elucidate the potential immunological mechanisms of protection. At the time of challenge infection, pigs vaccinated with LAV and KV vaccines had similar titers antibodies against the viral capsid (p72) protein, as measured by a commercial ELISA kit. However, utilizing a high-throughput assay to measure antibody reactivities against 29 viral structural proteins, we noticed distinct antibody profiles induced by LAV and KV vaccines. Particularly, the sera of LAV-vaccinated pigs reacted to 16 of 29 proteins, while the sera of KV-vaccinated pigs reacted to only 3 proteins. These findings provide valuable insights into the mechanisms of vaccine-induced protection against ASFV.

## 2. Materials and Methods

### 2.1. Cells, Viruses, and Control Antisera

HEK-293T cells (ATCC CRL-3216) were cultured in Dulbecco’s Modified Eagle Medium (DMEM) supplemented with 10% fetal bovine serum (FBS). Primary and immortalize PAMs (3D4/21 cell line, ATCC CRL-2843) were cultured in Roswell Park Memorial Institute 1640 (RPMI-1640) medium containing 10% FBS, as previously described [25]. The virulent virus VNUA-ASFV-05L1 (genotype II) was isolated from the spleen of a domestic pig in an ASF outbreak in northern Vietnam in 2020 [26]. A positive control antiserum containing antibodies against the Nluc protein was collected from a pig that was inoculated with a recombinant Senecavirus A expressing the Nluc gene.

### 2.2. Experimental Live-Attenuated Virus and Inactivated Virus Vaccine

The cell culture-adapted, live-attenuated virus was generated by successively passaging the virulent VNUA-ASFV-05L1 (genotype II) on primary PAM cultures for 65 passages and 3D4/21 cells for 55 passages. The attenuation and protective potential of the resulting virus VNUA-LAV-L2 passage 120 was reported previously [25]. In this study, the VNUA-LAV-L2 virus was further passaged on 3D4/21 cells for six additional passages. The virus was administered to pigs intramuscularly (i.m.) at a dose of 10^3^ hemadsorption dose 50 (HAD50) per animal.

The killed virus (KV) vaccine was generated based on the virulent VNUA-ASFV-05L1 cultured on primary PAMs. Briefly, the VNUA-ASFV-05L1 virus stock was inactivated using 1 mM binary ethyleneimine (BEI) for 24 h at 37 °C. Residual BEI was neutralized with 100 mM sodium thiosulphate for 2 h at 37 °C. The inactivation was verified by successively passaging the virus preparation for two passages on primary PAMs. The killed virus preparation was mixed with the oil-adjuvant MONTANIDE ISA-201 VG (Seppic, Courbevoie, France) at a ratio of 4:1 (*v*:*v*) in a way that each dose of the KV vaccine contained approximately 5 × 10^7^ HAD50 virus.

### 2.3. Animal Experiment

Thirty pigs, approximately 5 weeks old and seronegative for ASFV, were obtained from a local swine farm in Vietnam. The pigs were randomly assigned into three treatment groups and housed in three rooms at the Vietnam National University of Agriculture (VNUA) animal research facility. Group 1 served as a non-vaccinated (NV) control. Groups 2 and 3 were administered intramuscularly with the LAV or KV vaccines, respectively. The LAV vaccine was administered once at the beginning of the study. The KV vaccine was administered twice, with a three-week interval between vaccination.

Whole-blood samples with and without anticoagulant (ethylenediaminetetraacetic acid) were collected from all pigs before and weekly after immunization. The samples without an anticoagulant were used to collect serum samples to assess antibody responses. The samples with an anticoagulant were used to evaluate viremia after vaccination. 

At 42 days post-vaccination (dpv), the pigs were challenged by an intramuscular inoculation with 7 × 10^2^ HAD50 of the virulent virus VNUA-ASFV-05L1. The animals were monitored daily for 21 days. Whole blood samples were collected on different days post-challenge (dpc) to measure blood viral loads. Pigs were euthanized when reaching the humane endpoints.

### 2.4. Realtime PCR to Measure Viral Genomic DNA

Total DNA was extracted from whole blood samples using the MagMAX^TM^ CORE Nucleic acid purification kit (Life Sciences, New York, NY, USA) on the automated KingFisher™ Duo Prime DNA/RNA extraction system (ThermoFisher Scientific, Waltham, MA, USA). ASFV genomic DNA was quantified using a real-time PCR with the primers and probes specific to the viral p72 genes [27] and the Platinum™ Quantitative PCR SuperMix-UDG (Invitrogen, Waltham, MA, USA). Additionally, DNA was extracted from an ASFV stock with a known HAD50 titer. Subsequently, the viral DNA was diluted 10-fold serially, and real-time PCR was performed to generate a standard curve that allowed for the estimation of viral titers in the blood samples. Viral loads in blood were reported as log10 HAD50/mL equivalent. For statistical analysis, samples with undetectable levels of viral DNA were assigned a value of 0.

### 2.5. ELISA to Measure Antibody Responses

Anti-p72 and anti-p32 antibodies were measured using commercial ELISA kits. Anti-p72 antibodies were measured using the Ingezim PPA COMPAC ELISA (Ingenasa, Madrid, Spain), while anti-p32 antibodies were measured using the indirect ELISA (PrioCHECK™ African Swine Fever Virus Ab Kit, Applied Biosystems, San Francisco, CA, USA). Both assays were performed according to the manufacturer’s instructions.

### 2.6. Plasmid Construction and Protein Expression

The expression plasmid pCI-nluc.2 was generated by inserting a DNA fragment containing a flexible linker (GGGGSGGGGS) fused to the 5′ end of the *nanoluc luciferase* (*nluc*) gene (GenBank accession no. JQ437370.1) into the pCI vector (Promega, Madison, WI, USA). The resulting plasmid was linearized by PCR amplification using the primer pair pCI-Nluc-For (5′-GGTTCCGTATTCACCCTTGAG-3′) and pCI-Nluc-Rev (5′-GGTGGCTAGCCTATAGTGAGTCGTATTAAG-3′) to serve as a backbone for assembly of the ASFV genes.

Coding sequences of 68 structural genes of the ASFV Georgia 2007 (GenBank accession FR682468.2) were codon-optimized for optimal expression in human cells. Two stretches of 34 and 21 nucleotides overlapping with the CMV promoter and the flexible linker were incorporated into the 5′ and 3′ ends of each ASFV gene, respectively, to facilitate the cloning of the ASFV gene into the pCI-nluc.2 plasmid. The DNA fragments were chemically synthesized by Integrated DNA Technologies (Ann Arbor, MI, USA). Each DNA fragment was cloned into the pCI-nluc.2 using the HiFi DNA assembly master mix (New England BioLabs, Ipswich, MA, USA). This way, the ASF gene was cloned immediately downstream of the CMV promoter and in-frame with the flexible linker and the *nluc* sequence. As a result, the ASFV protein is separated from the *nluc* protein by a flexible linker. The resulting plasmids were sequenced to verify the authenticity of the ASFV genes.

To generate Nluc-tagged antigens, the plasmids were transfected into HEK-293T cells as previously described [28]. At 60 h post-transfection, the cells were collected and lysed in the RIPA lysis buffer (ThermoFisher Scientific, Rockford, IL, USA), supplemented with 1× protease inhibitor (Pierce Protease Inhibitor Tablet, EDTA-Free, ThermoFisher Scientific, Rockford, IL, USA). The cell lysates were centrifuged at 17,000× *g* for 10 min, and the supernatant was passed through a 0.45 mm filter to remove insoluble cell debris. The cell extracts were stored in small aliquots at −80 °C for future use.

### 2.7. Luciferase-Immunoprecipitation System

The luciferase-immunoprecipitation system (LIPS) was performed as previously described [28,29]. The overview of the LIPS assay is depicted in Figure 1. Briefly, test serum samples were diluted 1:40 in buffer A (50 mM Tris, pH 7.5, 100 mM NaCl, 5 mM MgCl2, 1% Triton X-100, Rohm & Haas, Philadelphia, PA, USA), and passed through a 0.45 μm filter to eliminate large aggregates. Fifty μL of each diluted sera were transferred to a well of a 96-well plate and mixed with 50 μL of the Nluc-tagged antigen extract containing approximately 10^7^ relative light units (RLUs). Each ASFV antigen was tested in duplicate. This way, the antibody reactivity of one serum sample can be tested with 47 ASFV antigens in one 96-well plate, together with a control antigen. The plate was incubated for 1 h at room temperature on a rocking platform. Ten μL of protein A Sepharose 4B (Invitrogen, Camarillo, CA, USA) pre-washed and diluted in 50 uL of buffer A was added to each well of the plate. After another 1 h incubation, the whole content from the 96-well plate was transferred to a 96-well filter HTS plate (EMD Millipore, Billerica, MA, USA) for washing on a vacuum manifold. After the final wash, Nano-Glo^®^ Luciferase substrate (Promega, Madison, WI, USA) was added to each well. The luminescence signal was measured using the SpectraMax L reader (Molecular Devices, San Jose, CA, USA). For each run, a swine serum sample containing antibodies against the Nluc protein was used as a positive control. Similarly, a pool of serum samples collected from ASFV-negative pigs was used as a negative control. The RLU of each test serum sample against the ASFV antigens was normalized by the RLU of the positive control antiserum that contained antibodies against the Nluc protein. Data were expressed as the ratios between the test serum sample’s RLUs and the negative control sample’s RLUs (S/N ratio).

### 2.8. Statistical Analysis

The log-rank test was used to analyze survival rates. One-way analysis of variance (ANOVA) was utilized for one-variable datasets, such as the p32-based ELISA titers of serum samples collected at 41 dpi or viral loads in blood samples at 7 dpc. Heatmaps and two-dimensional hierarchical clustering were created using the ComplexHeatmap package in R, with data being scaled by row. Spearman’s correlation coefficient was applied to assess the correlation between antibody intensity and viral loads.

## 3. Results

### 3.1. Virological and Serological Response after Vaccination

After vaccination, viral genomic DNA was not detected from pigs in the KV and NV groups throughout the 41-day observation period. On the other hand, low copies of viral genomic DNA, equivalent to 10^2^ HAD50/mL, were intermittently detected in the blood samples of three pigs in the LAV group between 14 and 35 dpv (Figure 2a). By 41 dpv, viral genomic DNA was no longer detected in the blood of pigs vaccinated with the LAV vaccine.

A commercial blocking ELISA kit designed to detect antibodies against the viral capsid protein p72 was employed to assess humoral immune responses after vaccination. Anti-p72 antibodies were not detected in pigs from the NV group (Figure 2b). Anti-p72 antibodies were first detected in eight of the 10 pigs in the LAV group at 14 dpv, and by 28 dpv, all pigs in the LAV group had anti-p72 antibodies (Figure 2b). Conversely, anti-p72 antibodies were only detected in pigs from the KV group at 28 dpv, corresponding to one week after the second vaccination. The intensity of anti-p72 antibodies gradually increased in both the LAV and KV groups. At 41 dpv, the levels of anti-p72 antibodies were similar between the LAV and KV groups. (Figure 2b).

### 3.2. Live-Attenuated Virus but Not Killed Virus Vaccine Conferred Protection against Lethal ASFV Challenge

After the challenge infection with the parental virulent ASFV strain, all pigs in the LAV group remained clinically healthy throughout the 21-day observation period. Conversely, pigs in the PBS and KV groups exhibited ASF-related signs at 7 dpc. The clinical signs progressed and reached the humane end-points at 10 dpc. By 16 dpc, seven pigs in the PBS group and eight pigs in the KV group were euthanized (Figure 3a). The remaining pigs in these two groups recovered and survived to the end of the 21-day observation period. 

At 3 dpc, viral genomic DNA was detected in five and six of the ten pigs in the PBS and KV groups, respectively. By 10 dpc, all pigs in the NV and KV groups had detectable viral genomic DNA in their blood (Figure 3b). Pigs in the LAV group displayed two distinct viremic profiles. Six pigs had no detectable viral genomic DNA in their blood until 10 dpc, thereafter, maintaining consistently low levels of viral genomic DNA (below 10^2^ HAD50/mL equivalent). Remarkably, one of these six pigs had undetectable levels of viral genomic DNA throughout the 21-day observation period (Figure 3b). The other four pigs in the LAV group had detectable viral genomic DNA in their blood by 7 dpc, and the levels of viral genomic DNA gradually increased, reaching the titers equivalent to 10^4^ and 10^5^ HAD50/mL equivalent. 

To facilitate a statistical comparison, the viral genomic DNA in blood samples collected at 7 dpc, when all pigs were still alive, were plotted (Figure 3c). Pigs in the LAV group had statistically lower titers of viral genomic DNA than those in the NV and KV groups. In contrast, the titers of viral genomic DNA from the KV group were not statistically different from that of the NV group.

### 3.3. Differential Antibody Responses between Pigs Vaccinated with a Live-Attenuated Virus Vaccine and Those Vaccinated with a Killed Virus Vaccine

We aimed to simultaneously measure antibody reactivities against 68 viral structural proteins, utilizing the LIPS assay. For this purpose, we cloned each individual structural ASFV gene in-frame to the 5′ end of the *nluc* gene in an expression plasmid. After that, these plasmids were transfected into HEK-293T cells to generate ASFV proteins fused to the N-terminus of the *nluc* reporter protein. Sixty-four structural genes were successfully expressed in HEK-293T cells. Since these ASFV proteins were fused in-frame to the *nluc* protein, it was expected that they should exhibit similar reactivity to the anti-nluc antibodies in the LIPS assay. However, our initial screening revealed that only 29 of 64 proteins reacted to a swine antiserum containing anti-nluc antibodies. Therefore, we focused on assessing the antibody reactivities of pigs vaccinated with either LAV or KV vaccines against these 29 ASFV structural proteins. 

A heatmap was constructed based on the intensity of antibody reactivities against the 29 ASFV proteins of the antisera collected at 41 dpv (Figure 4a). Generally, the sera of pigs within a treatment group exhibited a similar antibody profile. Eight sera collected from the LAV group exhibited robust antibody reactivities against most of the 29 proteins, thus forming a distinct cluster on the left side of the dendrogram. The remaining two sera of the LAV group did not react with most of the proteins and were clustered with the sera of the KV and NV groups. The sera from the KV group reacted moderately to a smaller group of ASFV proteins. Thus, they were grouped together at the midpoint of the dendrogram. The sera from the NV group did not react significantly with the 29 ASFV proteins, constituting a separate cluster on the right side of the dendrogram. Notably, two sera from the NV group exhibited moderate levels of non-specific reactivity with a few proteins. Consequently, they grouped together with sera from the KV group.

Next, we sought to identify immunogenic viral proteins recognized by the sera collected from the vaccinated pigs. We defined immunogenic viral proteins as those that the sera from vaccinated animals exhibited at least 4-fold higher intensity signals than the sera from the NV group and an adjusted *p*-value less than 0.05. Sera from the LAV-vaccinated pigs consistently reacted with 16 ASFV proteins, with the mean S/N ratios between 4 and 129. These 16 proteins are associated with various virion structures, including the outer envelope, inner envelope, core shell, and nucleoid. Conversely, sera from the KV-vaccinated group recognized only three viral inner envelope proteins, p12, p32, and p54 (Figure 4b). 

Since the sera from both the LAV and KV groups reacted with the p32 protein, we used a commercial ELISA designed to detect antibodies against the p32 to validate our LIPS assay results. All 10 sera of pigs in the NV group tested negative using the p32 ELISA (Figure 4c). Conversely, all sera of pigs in the LAV and KV groups had anti-p32 antibodies. Thus, the p32-ELISA results corroborate our LIPS results.

### 3.4. The Intensity of Antibody Binding to p12, p14, p15, p32, and pD205R Exhibited a Negative Correlation with the Viral Loads after Challenge Infection

After the challenge, pigs in the LAV group displayed two distinct viremic profiles. Six pigs had no detectable viral genomic DNA in their blood until 10 dpc and maintained consistently low levels of viral genomic DNA thereafter. The other four pigs had detectable viral genomic DNA in their blood at 7 dpc, with the viral genomic copy gradually increasing over time. Spearman’s correlation coefficient analysis was performed to study the relationship between the intensity of antibody reactivity and viral loads in the blood after the challenge infection. Five ASFV antigens, p12, p14, p15, p32, and pD205R, exhibited a negative correlation between the intensity of antibody reactivities and the magnitude of blood viral loads (Figure 5). Consequently, these antibody responses against these five antigens hold promise as potential markers of vaccine-induced protection.

## 4. Discussion

Antibodies are a component of the protective immunity against ASFV [12,13]. Therefore, identifying antibody responses against viral antigens associated with the protective immunity is critical for vaccine development and assessment of vaccine efficacy. LAV vaccines generally induce complete protection against genetically related virulent ASFV strains, while the KV vaccines often do not induce protection [19,20,21,24,30,31,32]. In light of this, we compare antibody profiles of pigs vaccinated with the LAV and KV vaccines to identify differential antibody responses that may be associates with vaccine-induced protection. Using an ELISA, we did not detect a significant difference in the intensity of anti-p72 antibodies between pigs vaccinated with the LAV and KV vaccines, indicating that the intensity of anti-p72 antibodies is not a reliable indicator of vaccine-induced protection. On the other hand, by utilizing a high throughput assay to assess antibody responses against 29 viral structural proteins, we observed distinct antibody profiles between pigs vaccinated with the LAV vaccine and those vaccinated with the KV vaccine. Specifically, KV-vaccinated pigs developed antibody responses against only three structural proteins, while LAV-vaccinated pigs exhibited antibody responses against 16 proteins. 

The distinct antibody profiles of pigs vaccinated with the KV and LAV vaccines might be attributed to the difference in quality and quantity of vaccine antigens. For the KV vaccine, the protein structure might be altered during the inactivation process, leading to a loss of immunogenicity. Different from the KV vaccine, the LAV can replicate in vaccinated pigs, leading to the amplification of the antigenic mass. In addition, the viral proteins are expressed inside the infected cells, allowing them to undergo proper post-translation modification and folding. These viral proteins are also processed by local lymphoid tissues, along with tissue-damaging (inflammatory) cytokines, making them highly immunogenic. As a result, the LAV vaccine induces antibody responses against a more extensive array of viral proteins than the KV vaccine.

For pigs that received the LAV vaccine, there was a negative correlation between the intensity of the antibody responses against five viral proteins, p12, p14, p15, p32, and pD205R, and the magnitude of viral loads in the blood after the challenge infection. The proteins p14 and p15 are products of the proteolytic processing of two polyproteins, pp220 and pp62, respectively [33,34]. These proteins localize at the core shell and are essential for the core assembly [34,35]. The protein pD205R, encoded by the *D205R* gene, is an RNA polymerase subunit 5 involved in the viral genome transcription [2]. The is not much information related to the immunogenicity of p14, p15, and pD205R. p12 is a viral attachment protein [36]. Anti-p12 antibodies are consistently detected in pigs infected with different ASFV strains [36,37]. However, swine sera containing anti-p12 antibodies do not exhibit virus-neutralization activities [36]. The protein p32, encoded by the *CP204L* gene, is expressed early after viral infection, and localizes predominantly in the cytoplasm of infected cells [38]. p32 is a highly immunogenic protein, and anti-p32 antibodies block virus infection of susceptible cells, mainly by preventing virus internalization [39]. Therefore, several p32-based ELISAs has been developed for serodiagnosis of ASFV infection [16,17]. Additionally, significant effort has been devoted to developing subunit vaccines containing the p32 protein. However, the protective efficacy of the p32-based vaccines is variable (reviewed in [22]). In the present study, all pigs vaccinated with the KV vaccine developed antibodies to p32, although the intensity of anti-p32 antibodies of pigs vaccinated with the KV vaccine was slightly lower than those vaccinated with the LAV vaccine, as measured by the ELISA assay (Figure 4b). It is unclear if antibodies against p32 alone are sufficient to confer protection. 

The LAV vaccine used in the present study did not demonstrate the same level of efficacy as reported in our initial study [25]. In the present study, nine of ten LAV-vaccinated pigs exhibited viral genomic DNA in their blood at multiple sampling dates, whereas in the previous study, only two of five vaccinated pigs exhibited low copies of viral genomic DNA at 5 days after challenge with the parental virulent ASFV strain. This variation may, in part, stem from differences in the vaccine virus’s ability to replicate in pigs. In the present study, viral genomic DNA was sporadically detected in three out of ten pigs after vaccination, whereas in the prior study, all five vaccinated pigs exhibited viral genomic DNA in their blood until 21 dpv. Thus, the LAV used in this study did not replicate in pigs as effectively as the virus used in the previous study. It is important to note that the LAV employed in this study was cultured for six additional passages on 3D4/21 cells compared to the virus used in the previous study. It is plausible that the virus may have been further attenuated after these six additional passages on this cell type.

We could only assess antibody reactivities against 29 of 68 viral structural proteins because many ASFV-Nluc fusion antigens were not recognized by a control swine serum containing anti-Nluc antibodies. In the context of the LIPS assay, the test antigens must be fused with a luciferase protein to enable the efficient detection of the antibody–antigen complexes [29]. Because viral proteins might contain a signal sequence at their N terminus to direct the nascent proteins to the appropriate cellular compartments for post-translation modifications [40], we decided to fuse the reporter gene *nluc* to the carboxy terminus of each ASFV gene to avoid interfering with the signal sequence functions. However, it has been reported in previous studies that antibody binding to an antigen-luciferase fusion protein might be affected by whether the luciferase protein is fused to the C-or N- termini of the antigens [41]. In future studies, we will assess antibody reactivity to two forms of ASFV-luciferase proteins, each carrying luciferases at either the C- or N-terminus. This approach will enhance the detection of antibody reactivities to the ASFV antigens.

## 5. Conclusions

The LAV vaccine protected pigs against a lethal challenge with a homologous virulent ASFV strain, while the KV vaccine did not confer the same protection. Pigs vaccinated with the LAV vaccine developed antibody responses against more viral proteins than those vaccinated with the KV vaccine. Within the group of pigs receiving the LAV vaccine, there was a negative correlation between the intensity of antibody reactivity against five ASFV antigens, specifically p12, p14, p15, p32, and pD205R and the levels of viral DNA in the blood of animals after a challenge infection with a virulent ASFV. These antigens warrant further evaluation as potential indicators of vaccine-induced protection.

## Figures and Tables

**Figure 1 vaccines-11-01687-f001:**
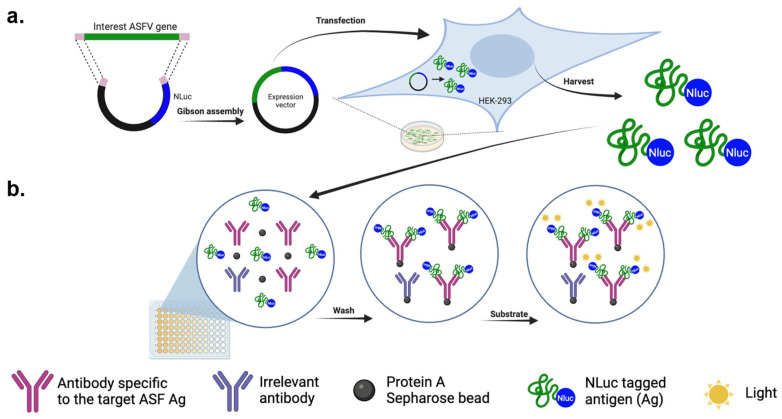
Overview of the LIPS assay. (**a**) The general approach to produce Nluc-tagged antigens (Ag). Individual structural genes containing overlapping sequences at both termini were chemically synthesized and cloned in-frame to the 5′- terminus of the *nluc* gene in the pCI-Nluc vector via Gibson assembly. The resulting plasmids were transfected into HEK 293-T cells to produce the Nluc-tagged ASFV antigens used for the LIPS. (**b**) Assessment of antibody binding to the NLuc-tagged antigens. Cell lysates containing Nluc-tagged antigens were incubated with test serum samples, together with protein A Sepharose beads, in a filter 96-well plate. When the test samples contained antibodies (IgG) specific to the Nluc-tagged antigen, antigen–antibody complexes formed and were captured by the protein A-Sepharose beads, thus being retained in the well. Unbound NLuc-tagged antigen was subsequently removed through washing. Following the addition of the luciferase substrate to the well, the Nluc-tagged antigen reacted with the substrate to produce light. The light intensity emitted by the Nluc-tagged antigen was directly proportional to the quantity of antigen-specific antibodies present in the test serum samples.

**Figure 2 vaccines-11-01687-f002:**
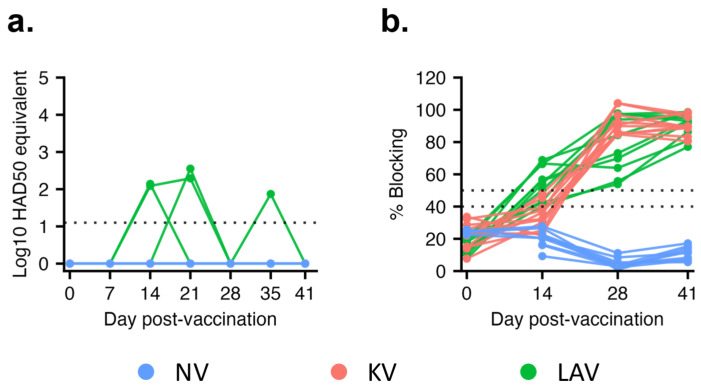
Viremia and antibody responses after vaccination. (**a**) Viremia quantification using a real-time PCR. To estimate the HAD50 equivalent in each sample, DNA samples extracted from a virus culture with a known HAD50 titer were utilized to establish a standard curve. Data are presented as log10 HAD50 equivalent. (**b**) Antibody responses measured by a blocking ELISA designed to detect anti-p72 antibodies. Two horizontal dotted lines at 40% and 50% indicate the cutoff values of the assay. Samples with % blocking equal to or above the upper cutoff were classified as positive, those equal to or below the lower cutoff were classified as negative, and those falling between the two lines were considered suspicious.

**Figure 3 vaccines-11-01687-f003:**
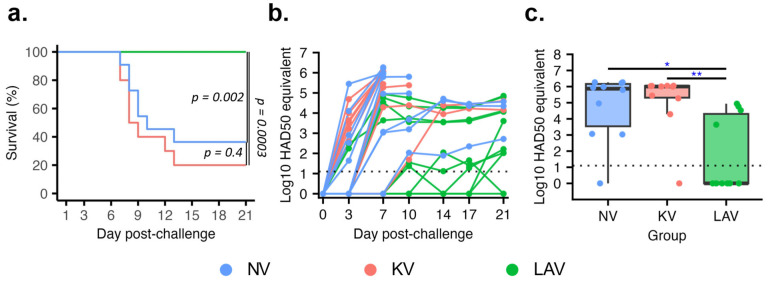
Protection outcomes after challenge infection with a virulent ASFV strain. (**a**) Survival rate after; (**b**) viral loads in blood samples quantified by a real-time PCR. Data are presented as log10 HAD50 equivalent as described in Figure 2. (**c**) Viral load in the blood samples collected at 7 dpc. The dotted horizontal lines in Figure 2b,c represent the limit of detection. * *p* ≤ 0.05; ** *p* ≤ 0.01.

**Figure 4 vaccines-11-01687-f004:**
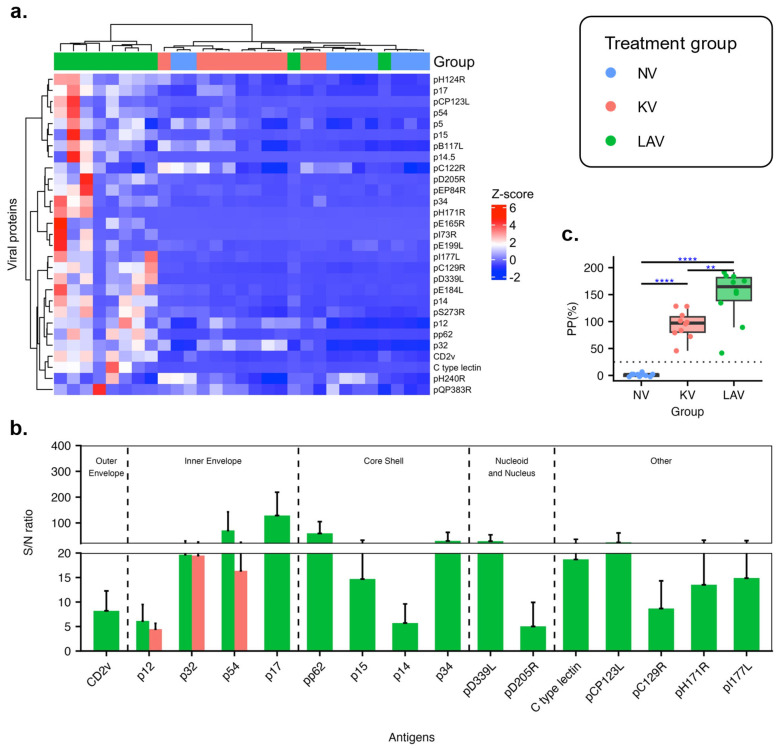
Antibody reactivities against a panel of 29 structural proteins. (**a**) Heatmap and two-dimension hierarchical clustering is based on the intensity of antibody reactivities of sera collected at 41 dpv against the 29 ASFV proteins. Each column represents a serum sample, and each row represents a viral protein. The heatmap color is coded based on the row Z-score. (**b**) The intensity of antibody reactivity against immunogenic viral proteins. Data are presented as the samples to negative (S/N) ratios. (**c**) Antibody responses measured by an indirect ELISA designed to detect anti-p32 antibodies. Data are expressed as percentage positivity (PP%). The dotted horizontal line indicates the assay cutoff. Test samples with a PP% value above this line were considered positive. ** *p* ≤ 0.01, **** *p* ≤ 0.0001.

**Figure 5 vaccines-11-01687-f005:**
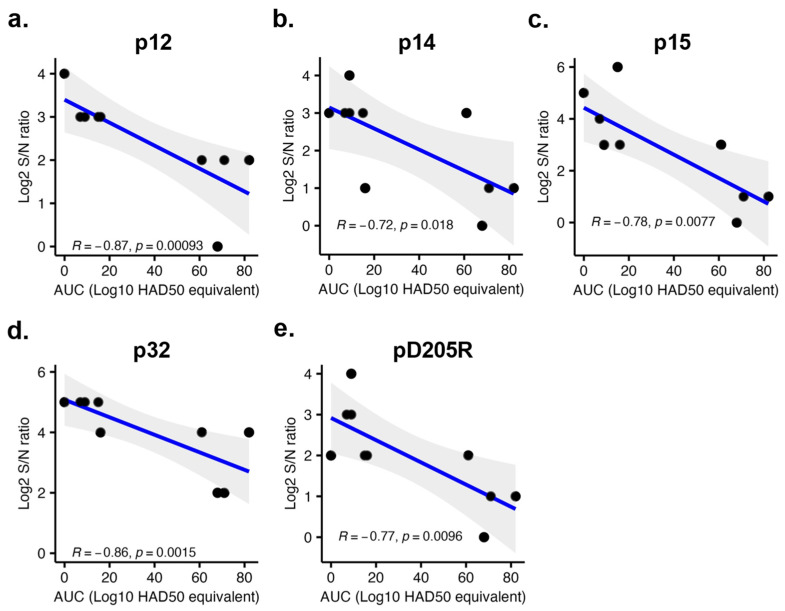
Spearman’s correlation coefficient analysis. The *y*-axis represents the log2 S/N ratio of serum samples collected at 41 dpi. The *x*-axis displays the Area Under the Curve (AUC) of viral loads following challenge infection. The viral antigen's identity is specified at the top of each graph.

## Data Availability

The article contains all the necessary data to support its main findings.

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
