# Peer review of "Comparative Analysis of Swine Antibody Responses following Vaccination with Live-Attenuated and Killed African Swine Fever Virus Vaccines"

_vaccines, 2023, doi:10.3390/vaccines11111687_

Round 1
Reviewer 1 Report
Comments and Suggestions for Authors
The manuscript submitted by Luong et al. entitled "Comparative analysis of swine antibody responses following vaccination with live-attenuated and killed African swine fever virus vaccines" aims to cmpare the antibody profiles of pigs vaccinated with two different vaccines in order to elucidate the potential immunological mechanisms of protection. At the time of challenge infection, animal had similar titers antibodies against the p72 but when authors used a high-throughput assay to measure antibodies against 29 ASFV proteins, distinct antibody profiles were found. Indeed, the pigs vaccinated with the LAV vaccine reacted to 16 of 29 proteins, while sera of the pigs vaccinated with the KV vaccine reacted to only 3 of 29 ASFV proteins.
The manuscript is well-written and provide some info about the immune response of pigs against two specific experimental vaccines. However, the number of the viral proteins is limited (29) and authors should discuss why those targets were selected. Addicionally, authors should also compare the production of Ab with results obtained from a bioinformatic screening in order to confirm if those proteins are signed as imunogenic. The images provided also have low definition that should be increase. In line 69, a recent review about ASFV vaccines should be added (10.1080/22221751.2022.2108342)
Author Response
1. The manuscript is well-written and provide some info about the immune response of pigs against two specific experimental vaccines.
Ans: We thank the reviewer for this supportive comment.
2. The number of the viral proteins is limited (29) and authors should discuss why those targets were selected.
Ans: In section 3.3, lines 300 – 312, we explained why we could only measure antibody responses against the 29 viral structural proteins.
3. Authors should also compare the production of Ab with results obtained from a bioinformatic screening in order to confirm if those proteins are signed as imunogenic.
Ans: In our opinion, bioinformatics tools are typically used to predict protein immunogenicity, not to confirm the experimental results. In all cases, the prediction obtained by bioinformatics prediction needs to be validated by actual immunological assays. In the present study, we utilized a validated method to measure antibody binding to the ASFV proteins. Thus, we do not see the value of using a bioinformatics tools to confirm our results.
4. The images provided also have low definition that should be increase.
Ans: We replaced the figures with higher resolution ones.
5. In line 69, a recent review about ASFV vaccines should be added (10.1080/22221751.2022.2108342)
Ans: The suggested reference is added.
Reviewer 2 Report
Comments and Suggestions for Authors
Comments on vaccines-2674566
Title: Comparative analysis of swine antibody responses following vaccination with live-attenuated and killed African swine fever virus vaccines
The present study by Hung Q. Luong et al. described the comparative analysis of antibody responses between the pigs vaccinated with a live-attenuated and a killed vaccine against African swine fever (ASFV). The work has something new and is potentially useful. However, several concerns should be addressed.
1. Why only the 29 ASFV structural proteins but not any ASFV non-structural proteins were included? This should be justified.
2. The 29 structural ASFV proteins should be labeled with a Flag or His tag to validate the expected expression.
3. The correlation of the antibodies with the pathological changes should be analyzed.
4. The invovlement of the cell-mediated immunity in the protection from ASFV challenge should not be ignored or underestimated.
5. The manuscript writing should be improved.
Comments on the Quality of English LanguageThe manuscript should be revised by native English speakers.
Author Response
The present study by Hung Q. Luong et al. described the comparative analysis of antibody responses between the pigs vaccinated with a live-attenuated and a killed vaccine against African swine fever (ASFV). The work has something new and is potentially useful. However, several concerns should be addressed.
1. Why only the 29 ASFV structural proteins but not any ASFV non-structural proteins were included? This should be justified.
Ans: Our overarching objective is to assess antibody responses against the entirety of ASFV proteon, consisting of approximately 170 proteins. We are pursuing this goal incrementally. In the current study, our primary aim is to evaluate antibody responses directed at the 68 structural proteins. To achieve this, we genetically fused the ASFV proteins with the nanoluciferase (Nluc) protein, employing these fusion constructs for measuring antibody binding. In theory, all 68 fusion proteins should exhibit affinity for the anti-Nluc antibody. However, approximately half of these fusion proteins were not pulled down by the anti-Nluc antibody in the LIPS assay and were consequently excluded from our analysis. As a result, we were only able to analyze the antibody response against these 29 of these proteins, as elaborated in Section 3.2, lines 300-312.
2. The 29 structural ASFV proteins should be labeled with a Flag or His tag to validate the expected expression.
Ans: We appreciate your suggestion. As clarified in point 1, all ASFV proteins are fused in-frame with Nluc, enabling their detection using the anti-Nluc antibody. Out of the 68 structural proteins, 64 were effectively expressed in HEK-293T cells. Nevertheless, only 29 of them were captured by the anti-Nluc antibody in the LIPS assay. This limitation is why we were only able to assess antibody responses against these 29 proteins.
3. The correlation of the antibodies with the pathological changes should be analyzed.
Ans: We appreciate your suggestion. Our collaborators have conducted a comprehensive analysis of the protective potential of the LAV vaccine candidate, as detailed in their preprint (doi: 10.20944/preprints202308.1896.v1). Their findings indicated the absence of significant pathological changes in pigs vaccinated with the LAV vaccine candidate, followed by a subsequently challenged with the virulent ASFV strain. Consequently, we did not consider this point when we designed this study.
4. The invovlement of the cell-mediated immunity in the protection from ASFV challenge should not be ignored or underestimated.
Ans: We concur with the reviewer's valuable suggestion. Enhancing our comprehension of host immune responses to ASFV infection requires a multifaceted approach. It's important to acknowledge that a comprehensive exploration of the host immune responses to ASFV vaccination (or natural infection). Other research groups have been actively investigating T-cell responses across the entire viral proteome. Meanwhile, our group primarily concentrates on studying humoral immune responses.
5. The manuscript writing should be improved.
Ans: We carefully reviewed the manuscript and corrected any grammatical errors.
Round 2
Reviewer 1 Report
Comments and Suggestions for Authors
The authors answered to all the questions raised by this reviewer.
Author Response
We thank the reviewer for constructive comments to our manuscript.
Reviewer 2 Report
Comments and Suggestions for Authors
The manuscript should be revised comprehensively.
See details in the attachment.

The English writing should be improved.
Author Response
Reviewer 1 pointed out that all the comments raised were effectively addressed, so no further edits were needed.
We appreciate the reviewer's dedication to reviewing our work. However, it appears that the reviewer made edits to the original manuscript, not the revised manuscript. We have already diligently revised our manuscript before resubmitting it last week. We tracked all the edits so the editors and reviewers could easily see the changes. Please see the revised manuscript that was submitted last week.